# Developing Wellbeing Through a Randomised Controlled Trial of a Martial Arts Based Intervention: An Alternative to the Anti-Bullying Approach

**DOI:** 10.3390/ijerph16010081

**Published:** 2018-12-29

**Authors:** Brian Moore, Stuart Woodcock, Dean Dudley

**Affiliations:** Department of Educational Studies, Faculty of Human Sciences, Macquarie University, North Ryde, NSW 2109, Australia; stuart.woodcock@mq.edu.au (S.W.); dean.dudley@mq.edu.au (D.D.)

**Keywords:** bullying, martial arts, mental health, resilience, self-efficacy, wellbeing

## Abstract

Anti-bullying policies and interventions are the main approach addressing bullying behaviours in Australian schools. However, the evidence supporting these approaches is inconsistent and its theoretical underpinning may be problematic. The current study examined the effects of a martial arts based psycho-social intervention on participants’ ratings of resilience and self-efficacy, delivered as a randomised controlled trial to 283 secondary school students. Results found a consistent pattern for strengths-based wellbeing outcomes. All measures relating to resilience and self-efficacy improved for the intervention group, whereas results declined for the control group. These findings suggest that a martial arts based psycho-social intervention may be an efficacious method of improving wellbeing outcomes including resilience and self-efficacy. The study proposes utilising alternatives to the anti-bullying approach and that interventions should be aimed towards helping individuals develop strengths and cope more effectively, which has specific relevance to bullying and more generalised importance to positive mental health.

## 1. Introduction

Bullying is a complex and controversial subject that receives frequent media and academic attention. However, there is no standard definition of bullying [1,2,3], and definitions inclusive of all bullying behaviours are difficult to establish [4]. Bullying behaviours can occur directly and indirectly; and include verbal, physical, and/or relational characteristics. Research suggests bullying can be defined as: (a) a type of aggression [5], (b) systematic and repeated [6], and (c) based upon an imbalance of power [7,8]. 

Anti-bullying policies and interventions are the main approach addressing bullying behaviours in Australian schools [9]. However, the evidence supporting these approaches is indeterminate (for example: [1,10,11]) and their theoretical underpinnings are problematic [4,5]. 

Examining Australian state and territory educational policies and related documents suggests there is a relatively consistent approach to bullying policy across Australia. While education is primarily a state and territory responsibility in Australia, there has been a collaborative approach to addressing bullying across Australian education jurisdictions. For example, the Bullying. No way! website [9] is a collaborative product of the Australian Government’s (AG) Department of Education, the state and territory government education departments, the National Catholic Education Commission, and the Independent Schools Council of Australia. This consistency is also due to school-based anti-bullying interventions being the main strategy addressing bullying behaviours [12]. Table 1 lists key policies across Australia, and Table 2 provides details of the characteristics of anti-bullying programs across Australian education jurisdictions. 

As indicated by Table 2, Australian anti-bullying policies have relatively consistent characteristics. Policies typically define bullying as: (a) involving the misuse of power in a relationship, (b) ongoing and repeated behaviour, and (c) involving behaviours intended to cause harm [9]. It is important to consider differences between the former (research oriented) and latter (policy oriented) definitions of bullying. First, the latter definition omits the notion that bullying is a form of aggression [5]. Second, while the issue of power is noted in the latter definition, the meaning of this is moderated by including the term misuse. Third, the latter definition includes a focus on intent and harm causation. While this issue is not excluded from the former definition, its use in the latter frames the way bullying is conceptualised in terms of victims and perpetrators of bullying. 

In response to these definitions, anti-bullying interventions typically fall into several categories. These include behavioural rule-based, non-punitive, social-emotional learning, and student intervention approaches. Policies typically mandate schools have official bullying or anti-bullying plans but suggest a flexible approach to implementation. Consequently, while policy and associated documents suggest a variety of strategies, it is difficult to know what anti-bullying actions are enacted at the school level.

### 1.1. The Failure of Anti-Bullying Policy

Anti-bullying literature frequently claims substantial empirical support to address bullying [1,25]. However, these claims should be viewed with caution. Case in point, a meta-analysis published by the Australian Government’s Attorney General’s Department reported that “with several exceptions … the outcomes from the evaluations reviewed were … positive in reducing overall bullying behaviour” [1] (p. 2). Despite Rigby’s [1] assertion that research shows bullying behaviours among school children can be “significantly reduced” (p. 1), this is not wholly supported by the evidence presented in the meta-analysis. Notably, of the 13 studies reviewed by the meta-analysis three studies reported no change and six studies cited increased bullying behaviour post-intervention.

Evidence provided by other meta-analyses vary considerably in the level of support they provide for the efficacy of anti-bullying approaches. Ttofi and Farrington [11], reported that anti-bullying programs reduced bullying behaviours by an average of 20–23 per cent; and Jiménez-Barbero, Ruiz-Hernández, Llor-Zaragoza, Pérez-García, and Llor-Esteban [26] reported that anti-bullying programs resulted in significant reductions in the frequency of bullying behaviours. By contrast, Merrell, Gueldner, Ross and Isava [10] reported mixed results regarding anti-bullying programs changing rates of bullying; and Smith, Schneider, Smith and Ananiadau [27] reported that only a small number of programs produced positive outcomes regarding bullying and victimisation. Despite claims of substantial empirical support, the evidence to date is largely inconclusive. 

The published meta-analyses raise another issue regarding anti-bullying policy and interventions to-date. It appears that the construct has become so broad that included studies contain almost any program that mentions bullying as a potential outcome variable. This leads to complicated methodologies and instruments being applied and it is therefore unsurprising that research to-date has not established what elements of anti-bullying strategies are crucial for reducing bullying behaviours [1]. Nonetheless, Farrington and Ttofi’s [12] meta-analysis suggested that several anti-bullying strategies appear to be associated with decreased bullying. The most important of these being disciplinary methods including using explicit consequences regarding bullying, followed by parent training and meetings, and the duration and intensity of bullying programs. 

### 1.2. An Evolutionary Perspective of Bullying

Bullying is a complex phenomenon that resists simple solutions [9]. Consequently, it is important to consider the narrative underpinning the anti-bullying approach to understand why the evidence regarding anti-bullying programs is inconclusive. As noted, anti-bullying policies typically omit the notion that bullying is a form of aggression [5] from their definitions. Aggression can be conceptualised as an exercise of power and relational power imbalances are critical to understanding bullying behaviours [28]. While anti-bullying literature proposed that bullying involves the abuse and exploitation of power [3,8], an evolutionary psychology perspective explores the use of power and rationalises why power imbalances exist.

Evolutionary psychology proposes that behaviours exist because they are adaptive [28]. Consequently, it is important to consider the adaptive nature of bullying, which appears to be a ubiquitous human behaviour [3,4]. Social dominance theory proposed that bullying can be used as a strategy to establish and maintain social dominance which facilitates access to valued resources, and can be understood in terms of the costs and benefits associated with using aggression [5]. It appears that many of the strategies used by the anti-bullying approach attempt to shift the relational balance of power without accounting for an evolutionary understanding of bullying behaviour. This occurs across behavioural, non-punitive, and student intervention anti-bullying strategies. 

Bystander intervention has been identified as a “core tenant” of anti-bullying programs [29] (pp. 31–32), and involves shifting bystanders to become “active defenders of bullied students” [25] (p. 8). This approach can be interpreted as attempting to change the relational balance of power and may be a conceptually optimistic strategy. Research regarding the bystander effect was developed due to the observation that bystanders are unlikely to help others [30]. Latane and Darley [30] suggested that there are many factors that inhibit helping and make bystander intervention less likely including bystanders noticing that something is happening, interpreting the event and deciding something is wrong, assuming personal responsibility for helping, and deciding how to help.

### 1.3. Resilience and Bullying

Resilience is a complex construct [31] that is often defined as the attainment of positive outcomes despite significant adversity, risk, or stress [32,33]. This is evident in the varying operational definitions of resilience which include hardiness, optimism, competence, self-esteem, social-skills, achievement, and absence of pathology in the face of adversity [34]. Resilience can be conceptualised as a multi-level construct that includes: (a) protective processes, (b) the interaction of protection and risks, and (c) conceptual tools used in predictive models [35]. Examining resilience in terms of protective factors offers a viable means of measuring the construct [36] and recent research has suggested specific resilience factors may have greater efficacy in developing resilience based approaches to bullying [37]. 

Generally, the research examining the relationship between resilience and bullying is limited. Positively, students who reported higher levels of resilience were found to be less likely to engage in bullying behaviours or be victims of bullying [38]. Resilience towards bullying behaviours appeared to improve when individuals could disclose their experiences to a peer or family member [39] which is supported by research finding family factors, such as warm relationships and positive home environments were associated with greater resilience to bullying [40]. Resilience programs that improved social skills also appeared to decrease bullying behaviours [41]. Greater resilience and self-efficacy to bullying was reported as correlating with students completing an adventure based anti-bullying initiative [42], however the study reported weak correlations (ranging from *r* = 0.25 to *r* = 0.26) to support its claims.

A recent study by Moore and Woodcock [28] examining the mental health impacts of bullying considered resilience from a factorial perspective and conceptualised resilience as incorporating a sense of mastery, relatedness to others, and emotional reactivity. The study found higher levels of mastery and relatedness were a protective factor regarding depression, lower levels of emotional reactivity were a protective factor regarding depression and anxiety, and that higher levels of emotional reactivity were associated with an increased likelihood of bullying and being a victim of bullying behaviours. In a follow-up study, Moore and Woodcock [37] examined the resilience sub-scales underpinning these factors and found a pattern of effect sizes that suggested the sub-scale factors optimism, trust, tolerance, sensitivity and impairment may be more important for developing efficacious interventions that promote resilience to the effects of bullying. 

### 1.4. Martial Arts Based Therapeutic Intervention and Developing Wellbeing

Martial arts are an ancient human behaviour which are often associated with promoting psychological benefits such as increased well-being, self-esteem and confidence [43]. The term martial arts are often used to describe many of the combat arts that developed in Eastern cultures [44] and martial arts participation exhibited substantial growth during the 20th century [45]. Perceptions of the martial arts are mixed [46]. The views range from martial arts promoting psychological benefits [43] to the assumption that martial arts training results in negative socialization processes such as increased aggression and hostility [46]. The focus of existing research has examined the physical aspects of martial arts training, such as health benefits and injuries resulting from martial arts practice [44]. Few studies have examined whether martial arts training can address mental health problems or promote mental health and wellbeing.

In a recent meta-analysis Moore, Dudley and Woodcock [47] reported that martial arts training had a positive effect on mental health outcomes. The analysis found that martial arts training had a small effect size regarding increasing wellbeing, and medium effect size regarding reducing internalising mental health issues, such as anxiety and depression. Several studies have reported martial arts training promoted characteristics associated with wellbeing. For example, a study examining a six-month taekwondo program reported increased self-esteem following the intervention [48]; while another found that participants reported higher self-concept compared to a comparison group after studying taekwondo for eight weeks [49]. Similarly, several studies reported that martial arts training reduced the symptoms associated with anxiety and depression. For example, karate students have been reported as being less prone to depression compared to reported norms for male college students [50]; and training in tai-chi reduced anxiety and depression compared to a non-treatment condition [51]. However, studies examining the psychological effects of martial arts training exhibit significant methodological problems that limit the generalisability of findings [46,52]. These include theoretical problems such as conceptual and definitional issues; and research design problems including a reliance on cross-sectional designs, small sample sizes, self-selection effects, limited use of follow-up measures, and not accounting for gender differences.

### 1.5. Bullying and a Martial Arts Based Psycho-Social Intervention

This study proposes an evolutionary perspective of bullying and considers the term bullying as a synonym for power imbalances in relationships. If it is assumed that bullying behaviours have an adaptive function, it may be optimistic to believe that bullying behaviours can be systemically modified. 

An alternative approach could involve more generalised interventions promoting mental health through developing strengths, wellbeing and the ability to cope. Martial arts training can be conceptualised as a sports-based mental health intervention, where sport provides “the hook” [53] (p. 124) with which to deliver psycho-social interventions. Given its emphasis on respect, self-regulation and health promotion, martial arts training may be efficacious alternative to the anti-bullying approach. However, the efficacy of martial arts based interventions has received little research attention [54]. This study examined a martial arts based intervention using a randomised controlled trial design, and considered whether this type of sports-based psycho-social approach has the potential to improve participants’ ratings of resilience and self-efficacy.

### 1.6. Theoretical Framework

The theoretical framework of this study is based on the bipartite martial arts model. Research examining the effects of martial arts training on mental health outcomes has typically used a bipartite model [55] which distinguishes between traditional and modern martial arts practice. This model assumes that the martial arts can be classified according to various criteria. The distinction between traditional and modern martial arts is based upon whether mediation, patterns practice and ethical and philosophical teachings are included (traditional martial arts) or excluded (modern martial arts) [56]. Traditional martial arts emphasize the non-aggressive aspects of martial arts, whereas modern martial arts typically emphasize competition and aggression [57]. The study is based on a traditional martial arts perspective.

### 1.7. Research Questions

Based on the literature review including identified gaps, methodological issues, and theoretical orientation; the following questions are proposed for investigation:How does participation in a 10-week martial arts based intervention affect mental health factors, including resilience and self-efficacy?; and,What is the relationship of mental health factors, including resilience and self-efficacy:
a)before a 10-week martial based intervention (pre-intervention)?b)immediately after a 10-week martial based intervention (post-intervention)? and,c)three months after a 10-week martial based intervention (follow-up)?

## 2. Materials and Methods 

### 2.1. Research Design 

The study was a 10-week secondary school-based intervention that has been evaluated using a randomised controlled trial. Ethics approval for the study was obtained from an Australian University Human Research Ethics Committee. The study was registered with the Australian and New Zealand Clinical Trials Registry (ACTRN12618001405202). Additionally, the study protocol was also reviewed externally by school psychologists employed by the NSW Department of Education.

The researchers conducted pre-intervention (baseline) assessments at participating schools after the initial recruitment process. Following pre-intervention assessments and randomisation, the intervention group received the intervention program after which post-intervention assessment was conducted. At the time of writing this article results from the planned 12-week post-intervention (follow-up) assessment were not available. The control group received the same intervention program after the first post-intervention assessment. The design, conduct and reporting of this study adhered to the Consolidation Standards of Reporting Trials (CONSORT) guidelines for a randomised controlled trial [58]. Participants and caregivers provided written informed consent.

### 2.2. Participants 

Two hundred and eighty-three (*N* = 283) students from five secondary schools in NSW, Australia were recruited to participate in the study. Participants had an age range of 12 to 14 years (*M* = 12.76, *SD* = 0.68) and were recruited from grade 7 (*n* = 192) and grade 8 (*n* = 91) at participating schools. One hundred and forty-three females and 136 males participated in the program. Four participants did not identify their biological sex. The socio-economic status of participants was reported as high (*n* = 70), high average (*n* = 85), low average (*n* = 49), and low (*n* = 78). The cultural background of the sample was predominantly reported as Australian (*n* = 251). Other reported cultural backgrounds included Aboriginal (*n* = 2), African (*n* = 1), Arabic (*n* = 1), Asian (*n* = 14), European (*n* = 3), and Pacific (*n* = 8). Two hundred and forty-one (*N* = 241) participants completed the post-intervention assessment. Of the 42 drop-outs from the study 20 participants chose to discontinue participation, 13 participants changed schools, and nine participants did not complete the assessment for unknown reasons. 

Power calculations were conducted to determine the sample size required to detect changes in resilience and self-efficacy outcomes resulting from martial arts training. Statistical power calculations assumed baseline-post-test expected effect size gains of *d* = 0.3, and were based on 90% power with alpha levels set at *p* < 0.05. The minimum completion sample size was calculated as *N* = 234. As participant drop-out rates of 20% are common in randomised controlled trials [59] the maximum proposed sample size was *N* = 293. 

### 2.3. Intervention Program 

The intervention was delivered in a face-to-face group format onsite at participating schools. The intervention dose was 10 × 50 min sessions, once per week for 10 weeks. Each intervention session included: (a)Psycho-education—based on facilitator guided group discussion. Topics included respect, goal-setting, self-concept and self-esteem, courage, resilience, bullying and peer pressure, self-care and caring for others, values, and, optimism and hope;(b)Warm up activities—basic exercises including jogging, star jumps, push ups, and sit ups;(c)Stretching activities—a variety of stretching activities were used during the program including hamstring stretch, triceps stretch, figure four stretch, butterfly stretch, lunging hip flexor stretch, knee to chest stretch, and standing quad stretch; and,(d)Technical martial arts practice—including stances, blocks, punching, and kicking. 

Additionally, the following activities were alternated during the program:(e)Patterns practice—a pattern is a choreographed sequence of movements consisting of combinations of blocks, punches and kicks, performed as though defending against imaginary opponents;(f)Sparring—an activity based on the tai-chi sticking hands exercise was included as an alternative to traditional martial arts sparring; and, (g)Meditation—based on breath focusing exercise. 

It should be noted that aggressive physical contact was not part of the intervention program. The intervention was delivered by a (1) registered psychologist with minimum 6 years of experience, and (2) 2nd Dan/level black-belt taekwondo instructor with minimum 5 years of experience.

### 2.4. Instruments 

Evaluation of the intervention program involved a variety of standardised psychometric instruments to report on mental health related outcomes. Instruments included the Strengths and Difficulties Questionnaire (SDQ) [60], Child and Youth Resilience Measure (CYRM-28) [61], and the Self-Efficacy Questionnaire for Children (SEQ-C) [62].

The SDQ is a 25-item instrument that measures various aspects of children’s and adolescents’ behaviour. The scale provided a total problems scale and five subscales including an emotional scale, conduct scale, hyperactivity-inattention scale, peer problems scale, and a prosocial behaviour scale. The English language self-report version of the scale was used which has a good internal consistency (0.76) [63] and good convergent validity. Items are scored on a 3-point Likert scale with 0 = not true, 1 = somewhat true, and 2 = certainly true. An example of SDQ items include: (1) I have one good friend or more, and (2) I am kind to younger children. 

The CYRM-28 is a 28-item instrument that measures various aspects of children’s and adolescents’ resilience. The scale provided a total resilience scale and three subscales including an individual capacities and resources scale, relationship with primary caregiver scale, and contextual factors scale. The self-report version of the scale was used which has good internal consistency (.66 to .81) [64] and good construct validity. Items are scored on a 5-point Likert scale with 0 = not at all, 1 = a little, 2 = somewhat, 3 = quite a bit, and 4 = a lot. An example of CYRM-28 items include: (1) I have people I look up to, and (2) I feel supported by my friends.

The SEQ-C is a 24-item instrument that measures various aspects of children’s and adolescents’ self-efficacy. The scale provided a total self-efficacy scale and three subscales including an academic self-efficacy scale, a social self-efficacy scale, and an emotional self-efficacy scale. The SEQ-C scales have good internal consistency (>0.70) [65]. Items are scored on a 5-point Likert scale with 0 = not at all, 1 = a little, 2 = somewhat, 3 = quite a bit, and 4 = very well. An example of SEQ-C items include: (1) How well can you become friends with other children? and (2) How well can you control your feeling?. 

### 2.5. Data Collection 

The sampling plan utilised a randomised controlled sample to generate data for the study. All eligible government or catholic secondary schools in an urban area of New South Wales, Australia (*n* = 140) were sent an initial email with an invitation to participate in the study. Schools that responded to the initial email were pooled and received a follow up call in random order from the project researchers to discuss their participation. The first five schools that demonstrated interest were then recruited into the study.

All students enrolled in grades 7 and 8 (target age range 12–14 years) at participating schools were invited to participate in the study. Participant and caregiver information and consent forms were provided to students. Two follow-up letters were subsequently sent at two week intervals. Students who responded to the invitation were pooled and randomly allocated into the study, or not included in the study. Concurrent martial arts training were exclusion criteria for participation in the study, however previous experience of martial arts training was not an exclusion criterion.

Data was collected at pre-intervention (baseline), post-intervention, and 12-week post-intervention (follow-up). Randomisation into intervention and control group occurred after pre-intervention assessments. A simple computer algorithm was used to randomly allocate participants into intervention or control groups. The researchers were blinded to this randomisation of intervention and control group allocations.

During data collection participants were withdrawn from regular classes in small groups (pre-intervention), or the groups in which they completed the program (post-intervention). This enabled the researchers to explain, monitor and provide assistance while participants completed the survey. Participants were reminded that the survey was confidential and that they could discontinue the survey at any point. Instructions given to participants included: (a) an explanation of rating scales, and (b) how to make corrections if necessary. Surveys were then given to participants. Upon completion participants placed the survey into a locked box to ensure confidentiality.

### 2.6. Data Analysis

Statistical analysis of the psychometric test data was conducted using Statistical Package for the Social Sciences version 25 (IBM SPSS Statistics, Chicago, IL, USA) and alpha levels were set at *p* < 0.05.

The collected psychometric test data was consolidated into subscale variables using factor analysis and the internal consistency of each variable was examined to determine reliability. This was completed for pre-intervention and post-intervention measures. Table 3 reports pre-intervention and post-intervention internal consistency for the SDQ, CYRM-28 and SEQ-C. During factor analysis the SDQ converged in four factors (the conduct subscale and attention subscale converged in one factor which was renamed behaviour problems). For the emotional problems subscale item 6 did not converge across pre-intervention and post-intervention factor analysis and was discarded from the study. Internal consistency for the emotional problems scale was good. For the behaviour problems subscale items 12 and 22 did not converge across pre-intervention and post-intervention factor analysis and were discarded from the study. Internal consistency for the behaviour problems scale was good. For the peer problems subscale items 12, 14 and 19 did not converge across pre-intervention and post-intervention factor analysis and were discarded from the study. Internal consistency for the peer problems scale was poor and the subscale was discarded from the study. All items converged for the prosocial behaviour subscale across pre-intervention and post-intervention factor analysis, however internal consistency for the prosocial behaviour scale was poor and the scale was discarded from the study. The internal consistency for the SDQ total problems scale was good across pre-intervention and post-intervention factor analysis. 

During factor analysis the CYRM-28 converged in three factors. For the individual capacities and resources subscale all factors converged across pre-intervention and post-intervention factor analysis, and internal consistency for the scale was good. For the relationship with primary caregiver subscale all factors converged across pre-intervention and post-intervention factor analysis, and internal consistency for the scale was good. For the contextual factors subscale all factors converged across pre-intervention and post-intervention factor analysis, and internal consistency for the scale was good. The internal consistency for the CYRM-28 total resilience scale was good across pre-intervention and post-intervention factor analysis. 

During factor analysis the SEQ-C converged in three factors. For the academic self-efficacy subscale all factors converged across pre-intervention and post-intervention factor analysis, and internal consistency for the scale was good. For the social self-efficacy subscale all factors converged across pre-intervention and post-intervention factor analysis, and internal consistency for the scale was good. For the emotional self-efficacy subscale item 8 did not converge across pre-intervention and post-intervention factor analysis and was discarded from the study. Internal consistency for the emotional self-efficacy scale was good. The internal consistency for the SEQ-C total self-efficacy scale was good across pre-intervention and post-intervention factor analysis.

Items to be included in the scale variables were added and computed to create composite scores for pre-intervention and post-intervention data. Multivariate analysis of variance (MANOVA), analysis of variance (ANOVA) and Chi-square analyses were used to examine test data. Interpretation of effect sizes reflected Cohen’s suggested small, medium, and large effect sizes, where *η*_p_^2^ sizes are equal to 0.10, 0.25, and 0.40 respectively [66].

## 3. Results

### 3.1. Behaviour and Emotion Problems 

The hypothesis that martial arts training would decrease total problems and related sub-factors was not supported. Using Pillai’s trace, martial arts training had no significant effect on participants’ emotional problems, *V* = 0.004, *F*(2, 238) = 0.52, *p* = 0.59, *η*_p_^2^ = 0.004; behavioural problems, *V* = 0.014, *F*(2, 237) = 1.65, *p* = 0.19, *η*_p_^2^ = 0.014; or total problems, *V* = 0.006, *F*(2, 237) = 0.75, *p* = 0.47, *η*_p_^2^ = 0.006. Means and standard deviations are summarized in Table 4.

### 3.2. Resilience 

The intervention improved levels of the overall resilience and resilience sub-factors. Means and standard deviations are summarized in Table 5. 

Individual capacities and resources: Using Pillai’s trace, there was a significant effect of the experimental condition on participants’ individual capacities and resources, *V* = 0.10, *F*(2, 238) = 13.35, *p* < 0.001, *η*_p_^2^ = 0.10. Separate univariate ANOVAs revealed no significant difference between the intervention and control condition pre-intervention, *F*(1, 239) = 1.64, *p* = 0.20, *η*_p_^2^ = 0.007; however there was a significant difference between the intervention and control condition post-intervention, *F*(1, 239) = 18.87, *p* < 0.001, *η*_p_^2^ = 0.07. 

Relationship with primary caregiver: There was a significant effect of the experimental condition on participants’ relationship with primary caregiver, *V* = 0.09, *F*(2, 238) = 11.59, *p* < 0.001, *η*_p_^2^ = 0.09. Separate univariate ANOVAs revealed no significant difference between the intervention and control condition pre-intervention, *F*(1, 239) = 1.06, *p* = 0.30, *η*_p_^2^ = 0.004; however there was a significant difference between the intervention and control condition post-intervention, *F*(1, 239) = 23.04, *p* < 0.001, *η*_p_^2^ = 0.09. 

Contextual factors: There was a significant effect of the experimental condition on participants’ contextual factors, *V* = 0.09, *F*(2, 238) = 11.94, *p* < 0.001, *η*_p_^2^ = 0.09. Separate univariate ANOVAs revealed no significant difference between the intervention and control condition pre-intervention, *F*(1, 239) = 1.54, *p* = 0.22, *η*_p_^2^ = 0.006; however there was a significant difference between the intervention and control condition post-intervention, *F*(1, 239) = 22.10, *p* < 0.001, *η*_p_^2^ = 0.09. 

Total resilience: There was a significant effect of the experimental condition on participants’ total resilience score, *V* = 0.14, *F*(2, 238) = 18.58, *p* < 0.001, *η*_p_^2^ = 0.14. Separate univariate ANOVAs revealed no significant difference between the intervention and control condition pre-intervention, *F*(1, 239) = 0.00, *p* = 0.98, *η*_p_^2^ = 0.000; however there was a significant difference between the intervention and control condition post-intervention *F*(1, 239) = 32.80, *p* < 0.001, *η*_p_^2^ = 0.12. 

### 3.3. Self-Efficacy

In terms of self-efficacy, the intervention improved levels of the overall self-efficacy and self-efficacy sub-factors. Means and standard deviations are summarized in Table 6. 

Academic self-efficacy: Using Pillai’s trace, there was a significant effect of the experimental condition on participants’ academic self-efficacy, *V* = 0.07, *F*(2, 238) = 9.19, *p* < 0.001, *η*_p_^2^ = 0.07. Separate univariate ANOVAs revealed no significant difference between the intervention and control condition pre-intervention, *F*(1, 239) = 0.67, *p* = 0.41, *η*_p_^2^ = 0.003; however there was a significant difference between the intervention and control condition post-intervention *F*(1, 239) = 17.98, *p* < 0.001, *η*_p_^2^ = 0.07.

Social self-efficacy: There was a significant effect of the experimental condition on participants’ social self-efficacy, *V* = 0.09, *F*(2, 238) = 12.14, *p* < 0.001, *η*_p_^2^ = 0.09. Separate univariate ANOVAs revealed a significant difference between the intervention and control condition pre-intervention, *F*(1, 239) = 6.24, *p* < 0.05, *η*_p_^2^ = 0.03; and there was a significant difference between the intervention and control condition post-intervention, *F*(1, 239) = 8.58, *p* < 0.01, *η*_p_^2^ = 0.04.

Emotional self-efficacy: There was a significant effect of the experimental condition on participants’ emotional self-efficacy, *V* = 0.09, *F*(2, 238) = 11.64, *p* < 0.001, *η*_p_^2^ = 0.09. Separate univariate ANOVAs revealed no significant difference between the intervention and control condition pre-intervention, *F*(1, 239) = 1.24, *p* = 0.27, *η*_p_^2^ = 0.005; however there was a significant difference between the intervention and control condition post-intervention, *F*(1, 239) = 15.95, *p* < 0.001, *η*_p_^2^ = 0.06.

Total self-efficacy: There was a significant effect of the experimental condition on participants’ total self-efficacy, *V* = 0.11, *F*(2, 238) = 14.94, *p* < 0.001, *η*_p_^2^ = 0.11. Separate univariate ANOVAs revealed no significant difference between the intervention and control condition pre-intervention, *F*(1, 239) = 1.48, *p* = 0.23, *η*_p_^2^ = 0.006; however there was a significant difference between the intervention and control condition post-intervention, *F*(1, 239) = 21.16, *p* < 0.001, *η*_p_^2^ = 0.08.

### 3.4. Post-Program Martial Arts Participation 

The intervention condition was asked whether they would like to continue practising martial arts after completing the program. Of the 125 participants who completed the intervention condition, 94 (75.2%) reported they would like to continue the program or participate in another martial art based program after the intervention. Chi square analysis of covariates found no significant differences for age, *χ*^2^ = 2.86, *p* > 0.05; grade, *χ*^2^ = 0.14, *p* > 0.05; biological sex, *χ*^2^ = 5.25, *p* > 0.05; cultural background *χ*^2^ = 4.85, *p* > 0.05; or socio-economic status *χ*^2^ = 7.83, *p* > 0.05.

## 4. Discussion

The results of the current study found a consistent pattern for wellbeing outcomes. All primary and secondary measures relating to resilience and self-efficacy improved for the intervention group and declined for the control group at statistically significant levels, which supports previous research [48,49]. The results provide valid and reliable evidence that psycho-social interventions based on a traditional martial arts model can be considered as an efficacious method of improving strengths and wellbeing outcomes. Interventions using this approach should promote an individual’s ability to cope with the effects of bullying. 

For resilience, the intervention had the greatest effect on total resilience which observed the largest effect size in the study. This is an important result given that students who report higher levels of resilience are less likely to become victims of bullying [38]. The resilience sub-factors relationship with primary caregivers and contextual factors exhibited the same effect size which was slightly smaller than for total resilience; and the resilience sub-factor individual capacities and resources exhibited the smallest resilience effect size resulting from the intervention program. The result regarding improved relationships with primary caregivers is particularly important given previous resilience research found that family factors including warm relationships and positive home environments are associated with greater resilience to bullying [40], and that resilience to bullying behaviours was improved when individuals could disclose their experiences to a family member [39]. The result that participants’ resilience improved for the intervention group and declined for the control group suggests the intervention may improve participants’ resilience to bullying.

A similar pattern was evident for self-efficacy insofar as the intervention had the greatest effect for total self-efficacy when compared to the self-efficacy sub-scales. Academic and emotional self-efficacy exhibited a similar effect size, which was slightly smaller than for total self-efficacy; while social self-efficacy exhibited the smallest self-efficacy effect size resulting from the intervention program. As the victims of bullying report lower self-efficacy compared to non-victims [37] the result that all self-efficacy scales improved for the intervention group and declined for the control group suggests the intervention may improve participants’ ability to cope with bullying.

While the study exhibited increased resilience and self-efficacy for the intervention group and decreases for the control group, comparison of effect sizes suggests that the intervention had a greater effect on resilience outcomes. To this end, it is notable that the intervention program had a greater effect regarding total resilience and two of the resilience subscales (relationship with primary caregivers and contextual factors); than for total self-efficacy. It is also interesting that the relationship with primary caregivers resilience sub-scale exhibited a larger effect size compared to the social self-efficacy sub-scale, as they are intuitively related scales. Further, it is important to note that while the specific wellbeing scales considered in the current study differ from previous research, the current results support previous findings which reported martial arts training improved wellbeing factors such as self-esteem [48] and self-concept [49].

Although the effect sizes for resilience and self-efficacy were in the small range, this is nonetheless an important result. First, the meta-analyses conducted by Moore, Dudley and Woodcock [47] found that martial arts training resulted in improvements to factors associated with wellbeing, but that observed effect sizes were small. Results from the current study are consistent with this. Second, as noted there is a clear pattern regarding all resilience and self-efficacy scales increasing for the intervention group and decreasing for the control group. Third, the study recruited a randomised sample from a normal population and the intervention program was a 10-week group program. Consequently, the intervention’s resilience and self-efficacy results should be viewed as promising. 

However, results from the current study do not support previous research findings [50,51] and a meta-analysis [47] suggesting that martial arts training had a significant impact on internalising mental health issues. The current study found no significant internalising mental health differences between the intervention and control groups post-intervention, which is intriguing in view of previous research and positive finding from the current study regarding wellbeing outcomes. 

It is encouraging that a large majority of the intervention group reported they would elect to continue martial arts participation post-program. Co-variate analysis was similarly encouraging, finding no differences for age, grade, biological sex, cultural background or socio-economic status regarding participants choosing to continue practicing martial arts. While the issue of self-selection effects cannot be entirely discounted, the result suggests the intervention program successfully engaged participants and may be applicable for a variety of different groups. 

### 4.1. Explanation of Intervention Effects 

The positive wellbeing outcomes found by the study are important, however explaining the casual factors associated with these effects is difficult. Resilience is a complex construct, and there are many elements in the intervention program incorporated from resilience literature that could result in improving resilience outcomes. These include empathic modelling, changing negative scripts, accepting participants for who they are and establishing realistic goals, encouraging participants to learn from mistakes, developing a social conscience, and encouraging problem solving and self-discipline [67]. Additionally, other social psychological constructs may have utility in explaining the intervention effects. For example, participants in the intervention program may have benefited from the development of an in-group identity [68] and the development of group superordinate goals [69]. From the current results the authors do not propose to isolate which factors may have been more important for developing greater resilience, however the above may be a useful conceptualisation towards this. These factors may also have utility in explaining participants’ improved self-efficacy resulting from the intervention program. 

The lack of observed effect post-intervention regarding internalising mental health issues may be explained by a variety of factors including scale issues, sample issues, as well as consideration of the null hypothesis. Scale issues may have impacted the study’s capacity to measure participants’ reports of internalising mental health issues. The SDQ exhibited problems regarding factor analysis and poor internal reliability, which resulted in several elements of the scale being discarded from the study. Consequently, it would be interesting to examine the effects of martial arts training regarding internalising mental health using an alternate measure. Sample bias may also have contributed to the study not detecting changes regarding internalising mental health. Given that the sample was randomly recruited from a normal population, the current study may not have had the same capacity to detect changes to participants’ internalising mental health issues compared to previous research that recruited from targeted pathological samples. For example, Trulson’s [48] study recruited from a sample of males diagnosed with behaviour disorders. Finally, it is important to consider the null hypothesis. It is possible that martial arts based training does not have a significant effect on internalising mental health outcomes. However, the authors propose that the noted scale and sample issues require further investigation before making conclusions regarding the null hypothesis.

### 4.2. Limitations

Several issues may limit the results of the current study. First, while the study has addressed many of limitations evident in prior research, the current study did not obtain third party corroboration of self-report measures. Inclusion of third party report measures should be considered in future research. Longer term effects of martial art training on mental health outcomes are unclear, given that follow-up data was not available at the time of writing this article. Finally, the issue of psycho-education versus martial arts training may be a confound in the current study. Psycho-education and martial arts training were presented as part of a single intervention program, hence it is not possible to separate their effects. However, it is arguable that combined psycho-education and martial arts training is consistent with the practice of many traditional martial arts, hence this issue may only have superficial significance. 

### 4.3. Implications Regarding Interventions to Reduce Bullying 

The current study’s results have multiple implications to policy, practice and future research regarding bullying. Bullying behaviours are difficult to address, and the results attributed to the anti-bullying approach are inconsistent. Despite this, the anti-bullying approach maintains a dominant position informing educational policy regarding bullying. Arguably, the theoretical underpinning informing educational policy should be modified to account for other explanations of bullying behaviour, such as evolutionary perspectives on power and bullying. This should lead to questions regarding whether the aims of the anti-bullying approach (namely: no bullying) are achievable. The current study proposes an alternate approach to educational policy and suggests that instead of focusing resources towards eliminating bullying behaviours, policy should focus on promoting mental health through developing wellbeing. Results from the current study suggest that martial arts based psycho-social interventions have the potential to improve participants’ strengths, wellbeing and ability to cope; which is important to consider in terms of educational policy regarding bullying. 

The study has significant implications to practice regarding bullying behaviour. The institutional failure of the anti-bullying approach is potentially damaging to individuals. It is important to consider the impact and message understood by students when they continue to experience bullying behaviours despite the explicit anti-bullying policies and strategies used by schools. Further, because schools combine a wide variety of anti-bullying strategies flexibly, it is difficult to know what anti-bullying actions occur at the school level, and concerns should exist regarding whether the specific combinations used by schools are empirically supported. Given the inconsistent results attributed to the anti-bullying approach, alternate practices regarding bullying behaviour should be considered. Specifically, the current results have implications for practice by suggesting that intervention programs that develop strengths, including resilience and self-efficacy, have the potential to improve participants’ sense of wellbeing and ability to cope with bullying. 

The current study employed a robust design and rigorous evaluation, and suggests a promising approach to addressing bullying behaviours through improving wellbeing and mental health. The current results have a variety of implications for future research including sample considerations, methodological variations and developing a teacher professional learning program based on the study. Future research should examine the intervention program’s effects on different population samples. Consideration should be given to implementing the intervention as a universal program with primary school students (ages 10–11); targeting specific populations such as participants diagnosed with (a) mental health issues such as anxiety and depression, (b) Autism Spectrum Disorder, (c) intellectual disability, and (d) behavioural disorders such as Oppositional Defiance Disorder, and participants who have experienced violent incidents including domestic and sexual violence; and extending the study to international settings to examine the intervention’s efficacy in different cultural milieus. Future research should incorporate methodological changes including utilising third party measures to corroborate self-report measures, using alternate measures of mental health pathology such as the Kessler 10 Psychological Distress Scale [70], including a qualitative approach to supplement quantitative information, and varying the longitudinal parameters of the study by extending the follow-up period and lengthening the intervention program (e.g., 3/6/12 month interventions). Finally, future research should develop a professional learning program for teachers to facilitate similar programs which could be embedded within a physical education curriculum or welfare/pastoral care curriculum. Such a program would require piloting and ongoing measurement to ensure program efficacy.

## 5. Conclusions

Despite inconsistent evidence, the anti-bullying approach dominates educational policy and practice regarding bullying behaviours. While bullying is a complex phenomenon, the inconsistent evidence regarding the anti-bullying approach may be explained by problems with its theoretical underpinning. The authors propose the primary aim regarding bullying behaviours should change from attempted reduction of the behaviour, to improving individual ability to cope with the effects of bullying. The current study offers a strengths-based method of achieving this which is consistent with previous research. The results suggest that psycho-social interventions based on a traditional martial arts model can successfully promote wellbeing characteristics such as resilience and self-efficacy. Interventions that promote these strengths can help individuals develop better mental health and should be a focus of efforts to address the effects of bullying. 

## Figures and Tables

**Table 1 ijerph-16-00081-t001:** Australian educational bullying policy and associated documents.

Education Jurisdiction	Bullying Policy Accessible Online	Policy Document Title
ACT (Territory)	Yes	Safe and supportive schools policy [13]
AG (Federal)	No ^a^	Disability Discrimination Act 1992 [14] Human Rights and Equal Opportunity Commission (HREOC) Act 1986 [15] Racial Discrimination Act 1975 [16] Racial Hatred Act 1995 [17] Sex Discrimination Act 1984 [18]
NSW (State)	Yes	Bullying of students—Prevention and response policy [19]
NT (Territory)	Limited	Health and wellbeing of students: Bullying, cyberbullying and cybersafety [20]
QLD (State)	Yes	Preventing bullying and violence [21]
SA (State)	Limited	Keeping children safe from bullying [22]
TAS (State)	No	n/a
VIC (State)	Yes	School Policy—Bullying [23]
WA (State)	Yes	Guidelines for preventing and managing bullying in schools [24]

Note. ACT: Australian Capital Territory, AG: Australian Government, NSW: New South Wales, NT: Northern Territory, QLD: Queensland, SA: South Australia, TAS: Tasmania, VIC: Victoria, WA: Western Australia. ^a^ While there is no specific Federal policy regarding school bullying the listed Federal legislation is relevant to this area.

**Table 2 ijerph-16-00081-t002:** Characteristics of Australian anti-bullying programs based on policy and associated documents.

Education Jurisdiction	Defines Bullying	Mandates Bullying Plan	Whole School	Preventative Strategies	Awareness Raising	Staff Training	Engage Carers	Behaviour Sanctions	Non-Punitive	Social-Emotional	Bystander Intervention
ACT (Territory)	√	√	√	√	√	√	√	√	√	√	x
AG (Federal)	√	n/a	√	√	√	√	√	√	√	√	√
NSW (State)	√	√	√	√	√	√	√	√	√	√	√
NT (Territory)	√	x	x	x	x	x	x	x	x	x	x
QLD (State)	√	√	√	√	√	√	√	x	√	√	√
SA (State)	√	√	√	√	x	√	√	x	x	x	x
TAS (State)	n/a	n/a	n/a	n/a	n/a	n/a	n/a	n/a	n/a	n/a	n/a
VIC (State)	√	√	√	√	√	√	√	x	√	√	√
WA (State)	√	√	√	√	√	√	√	x	√	√	x

Note. ACT: Australian Capital Territory, AG: Australian Government, NSW: New South Wales, NT: Northern Territory, QLD: Queensland, SA: South Australia, TAS: Tasmania, VIC: Victoria, WA: Western Australia, √ = yes, x = not stated, n/a = not applicable.

**Table 3 ijerph-16-00081-t003:** Internal consistency for SDQ, CYRM-28 and SEQ-C across pre-intervention and post-intervention measures.

Measure	Scale	Pre-Intervention α	Post-Intervention α
SDQ	Emotional problems	0.72	0.73
	Behaviour problems	0.81	0.81
	Peer problems	0.18	0.12
	Prosocial behaviour	0.50	0.45
	Total problems	0.72	0.76
CYRM-28	Individual capacities and resources	0.85	0.89
	Relationship with primary carer	0.80	0.81
	Contextual factors	0.73	0.74
	Total resilience	0.89	0.91
SEQ-C	Academic self-efficacy	0.83	0.84
	Social self-efficacy	0.78	0.82
	Emotional self-efficacy	0.81	0.85
	Total self-efficacy	0.89	0.91

**Table 4 ijerph-16-00081-t004:** Means and standard deviations for SDQ strengths and difficulties scales by experimental condition.

Scale	Condition	Baseline		Post-Test	
		Mean	SD	Mean	SD
Emotional difficulties	Intervention Control	0.69 0.74	0.51 0.51	0.66 0.73	0.47 0.51
Behavioural difficulties	Intervention Control	0.77 0.78	0.42 0.48	0.77 0.68	0.45 0.43
Total difficulties	Intervention Control	0.82 0.81	0.23 0.24	0.81 0.78	0.24 0.25

**Table 5 ijerph-16-00081-t005:** Means and standard deviations for CYRM-28 resilience scales by experimental condition.

Scale	Condition	Baseline		Post-Test	
		Mean	SD	Mean	SD
Individual capacities and resources	Intervention Control	2.96 3.06	0.60 0.59	3.10 2.75	0.53 0.69
Relationship with primary carer	Intervention Control	3.14 3.05	0.68 0.74	3.17 2.73	0.65 0.79
Contextual factors	Intervention Control	2.51 2.38	0.76 0.77	2.62 2.18	0.72 0.71
Total resilience	Intervention Control	2.90 2.90	0.51 0.56	3.01 2.62	0.45 0.61

**Table 6 ijerph-16-00081-t006:** Means and standard deviations for SEQ-C self-efficacy scales by experimental condition.

Scale	Condition	Baseline		Post-Test	
		Mean	SD	Mean	SD
Academic self-efficacy	Intervention Control	2.56 2.48	0.71 0.73	2.77 2.41	0.64 0.69
Social self-efficacy	Intervention Control	2.57 2.78	0.74 0.57	2.84 2.60	0.62 0.67
Emotional self-efficacy	Intervention Control	2.23 2.34	0.80 0.73	2.63 2.25	0.70 0.79
Total self-efficacy	Intervention Control	2.45 2.54	0.60 0.52	2.75 2.42	0.52 0.59

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
