# Peer review of "Developing Wellbeing Through a Randomised Controlled Trial of a Martial Arts Based Intervention: An Alternative to the Anti-Bullying Approach"

_ijerph, 2018, doi:10.3390/ijerph16010081_

Round 1
Reviewer 1 Report
A novel contribution which is very much needed in fields of sports psychology, sports science, mens' health and health and wellbeing in particular. This has potential to be expanded in many ways in future, so is a great platform to pivot from. A very good read.
Lines 438-439, repeated in lines 482-484: How are these two lines different, repetition perhaps? If not, then will need to be clearer
I would forgo using the term 'cyber-bullying' (see 4.3). bullying and cyber-bullying are two 'different' things, and there hasn't been any rationale given as to why and how it should now be included.
Perhaps including a statement in the discussion of the benefits it will also have for those students who are/have been victims of sexual violence. Possibly suggesting this would also be a future avenue for further investigation.
Overall a very good study and worthy of publication after a very close proof-read.
Author Response
Response to Reviewer 1 Comments
Point 1: A novel contribution which is very much needed in fields of sports psychology, sports science, mens' health and health and wellbeing in particular. This has potential to be expanded in many ways in future, so is a great platform to pivot from. A very good read.

Response 1: Thank you.
Point 2: Lines 438-439, repeated in lines 482-484: How are these two lines different, repetition perhaps? If not, then will need to be clearer
Response 2: Agreed, this is repetitive. We have deleted lines 438-439.
Point 3: I would forgo using the term 'cyber-bullying' (see 4.3). bullying and cyber-bullying are two 'different' things, and there hasn't been any rationale given as to why and how it should now be included.
Response 3: Agreed. We have deleted the term 'cyber-bullying' from section 4.3.
Point 4: Perhaps including a statement in the discussion of the benefits it will also have for those students who are/have been victims of sexual violence. Possibly suggesting this would also be a future avenue for further investigation.
Response 4: This is a good suggestion for future research. Thank you. We have added a statement regarding this (line 577-578).
Point 5: Overall a very good study and worthy of publication after a very close proof-read.
Response 5: Thank you.
Reviewer 2 Report
I thoroughly enjoyed reading this well written and well-argued paper presenting a randomised controlled trial of a martial arts approach to reducing bullying in Australian schools.
I do not have any major issues with the paper but would mention two things:
1. Given the thorough and effective review of policy and research in the field, there is a slight degree of needless repetition at the top of section 1.5.
2. In the materials and methods section there is a mixture of tense in the reporting, which moves around past and present tense at the beginning of this section. I suggest the study is presented entirely in past tense.
Author Response
Response to Reviewer 2 Comments
Point 1: I thoroughly enjoyed reading this well written and well-argued paper presenting a randomised controlled trial of a martial arts approach to reducing bullying in Australian schools. I do not have any major issues with the paper but would mention two things:

Response 1: Thank you.
Point 2: Given the thorough and effective review of policy and research in the field, there is a slight degree of needless repetition at the top of section 1.5.
Response 2: Agreed. We have deleted lines 180-184.
Point 3: In the materials and methods section there is a mixture of tense in the reporting, which moves around past and present tense at the beginning of this section. I suggest the study is presented entirely in past tense.
Response 3: Agreed. The tense at the beginning of this section is not consistent. We have revised so that tense is presented in past tense.